# SYSTEMATIC EXPLORATION SUPERVISION ENABLES SCALING BEYOND TRAINING COMPLEXITY

## ABSTRACT

Language models trained on input–output pairs or linear Chain-of-Thought (CoT) traces often fail when task complexity at test time exceeds the regime seen during training; reinforcement learning methods can help but suffer from cold-start brittleness when base accuracy is low. We introduce *Systematic Exploration Supervision (SES)*, a process-level supervision framework that verbalizes *complete multi-branch search traces* (sampling alternatives, propagating outcomes, and backtracking to extract a solution) rather than a single reasoning chain. In textualized Gridworld, SES preserves 76.5% success when scaling from 10×10 training environments to unseen 20×20 grids (vs. 19.0% for standard supervised fine-tuning, 26.0% for inference-time Tree-of-Thought, and 6.0% for GRPO). We further extend SES to open domains via a bootstrapped trace construction procedure that guarantees inclusion of at least one valid solution while adding diverse, reward-prioritized alternatives. Results show substantial improvements on combinatorial reasoning (Game of 24: 47% vs. 17% best baseline) and competitive performance on logical reasoning (ProntoQA: 100%), with task-dependent effectiveness patterns. We demonstrate that SES behavior cannot be induced with few-shot prompting alone, even with sophisticated models like GPT-o1, suggesting *in-weight algorithmic policy* acquisition. Remarkably, our approach achieves 14× parameter efficiency, with a 0.5B model outperforming 7B baselines. We characterize when SES is advantageous (large branching factors, low base competence, scaling demands) and discuss limitations (token length inflation, effectiveness when base models are already competent). Our findings highlight full-search verbalization as a simple, offline alternative to inference-time search or costly RL for scaling systematic reasoning.

## 1 INTRODUCTION

Recent advancements in language models have demonstrated remarkable proficiency across diverse tasks (Brown et al., 2020; Touvron et al., 2023; Chowdhery et al., 2023; Chen et al., 2021), yet they remain brittle on *scaling* variants of structured, long-horizon problems. Controlled studies (Einstein's puzzle, multi-digit arithmetic (Dziri et al., 2024), Blocksworld (Valmeekam et al., 2022)) show sharp failure once instance complexity exceeds training conditions, even for GPT-4 (OpenAI et al., 2024). This gap separates memorized pattern replay from reusable problem-solving *procedures*.

Consider a model trained only on 10×10 grid navigation that must act in a 20×20 grid. The transition dynamics and optimality criteria are unchanged, yet standard supervised fine-tuning (SFT) collapses. We hypothesize the missing ingredient is not more examples, but **explicit supervision of exploration structure**, i.e. teaching how to enumerate, evaluate, and prune alternatives before committing.

Existing paradigms have complementary but limited strengths: (i) **SFT / linear CoT** supervise a single path and fail to endow branching discipline (Wei et al., 2023; Kim et al., 2023); (ii) **Inference-time search** (e.g., Tree-of-Thought) performs exploration externally, incurring runtime cost and sometimes weak generalization; (iii) **RL methods** (e.g., GRPO (Shao et al., 2024)) can refine policies but struggle under low initial success (cold start) and sparse rewards. Our goal is an *offline* alternative that directly teaches an internalized search operator.

We propose *Systematic Exploration Supervision (SES)*, which verbalizes *complete multi-branch search traces* (not just the winning path) so the model learns a reusable exploration procedure.

SES comprises two complementary instantiations: (1) **Direct algorithmic supervision** (when a canonical search procedure such as BFS/DFS is available), and (2) **Bootstrapped search discovery** that constructs synthetic trees containing at least one verified solution plus diverse, reward-prioritized alternatives when no explicit algorithm is given. Our contributions:

- **Framework**: A unified formulation of Systematic Exploration Supervision (SES) that verbalizes sampling, propagation, and backtracking across branches.
- **Scaling result**: SES sustains 76.5% success on $20 \times 20$ grids after training only within $10 \times 10$ grid (vs. 19.0% SFT, 26.0% ToT, 6.0% GRPO).
- **Parameter efficiency**: Remarkable 14× parameter efficiency where a 0.5B SES model (68%) outperforms 7B baseline models, suggesting algorithmic competencies independent of scale.
- **Bootstrapped extension**: A tree construction procedure guaranteeing at least one valid solution while adding reward-ordered, deduplicated alternatives for open-domain reasoning.
- **Unpromptable behavior**: SES-style reasoning cannot be induced through few-shot prompting alone, even with sophisticated models like GPT-o1, indicating in-weight algorithmic policy acquisition.
- **Task characterization**: Empirical conditions where SES excels (large branching factor, low base accuracy, complexity extrapolation) vs. regimes where conventional methods remain preferable.

## 2 BACKGROUND AND PROBLEM FORMULATION

Our work builds on process supervision (Uesato et al., 2022; Lightman et al., 2023), which provides intermediate guidance during reasoning by supervising step-by-step solutions rather than just final answers. However, existing process supervision focuses on linear reasoning chains that follow a single solution path. Our contribution extends process supervision to systematic exploration—teaching models how to structurally explore multiple solution paths, evaluate alternatives, and extract optimal solutions through explicit search procedures.

**The Central Challenge: Training vs. Test Complexity Gap.** Current language models exhibit brittle scaling behavior when test-time complexity exceeds training distributions (Dziri et al., 2024; Valmeekam et al., 2022). This brittleness manifests across three key dimensions: (1) **Structural complexity:** longer reasoning chains, deeper search trees, and more decision points; (2) **Environmental complexity:** larger state spaces, increased branching factors, and novel configurations; (3) **Procedural complexity:** tasks requiring systematic exploration rather than pattern matching. Traditional approaches fail because they teach *what* decisions to make rather than *how* to organize exploration when facing unprecedented complexity.

**Systematic Exploration vs. Alternative Paradigms.** Our approach addresses fundamental limitations in existing reasoning paradigms. **Supervised fine-tuning** on linear Chain-of-Thought traces teaches individual solution paths but fails to capture the systematic exploration process underlying human problem-solving. **Reinforcement learning methods** can refine policies through environmental feedback but suffer from cold-start problems when base competence is low and reward signals are sparse. **Inference-time search** methods like Tree-of-Thought perform exploration externally, incurring computational overhead and sometimes achieving weak generalization to novel environments. We introduce Systematic Exploration Supervision (SES) as an offline alternative that directly teaches internalized search procedures. We briefly survey related work (detailed exposition in Appendix A).

**Complexity-Based Evaluation Framework.** We use textualized Gridworld as our primary controlled evaluation environment because it enables systematic complexity scaling while minimizing confounding factors from domain knowledge. For any environment instance $E$, we define policy-wise complexity under model $\pi_\theta$ as the negative log-likelihood assigned to the optimal action sequence $\tau^* = (s_1, a_1, \ldots, s_L, a_L)$:

$$C(E, \pi_\theta) = -\sum_{t=1}^{L} \log \pi_\theta(a_t \mid s_t) \tag{1}$$

This complexity metric increases with branching factor and path length even when underlying transition dynamics remain unchanged, providing a principled measure of extrapolation difficulty that correlates with model performance degradation.

**Paper Organization and Contributions.** We first formalize our Systematic Exploration Supervision framework (Section 3) with two complementary instantiations: direct algorithmic supervision for domains with known search procedures, and bootstrapped search discovery for open-ended reasoning tasks. We evaluate SES across four different tasks(Section 4). First, we demonstrate SES in controlled environments (Section 5) to see complexity scalability, and then extend to open domains (Section 6) showing task-dependent improvements. We analyze settings when SES excels along with the conclusion (Section 7).

# 3 SYSTEMATIC EXPLORATION SUPERVISION FRAMEWORK

## 3.1 CORE PRINCIPLE: COMPLETE SEARCH PROCESS SUPERVISION

The central insight is that scaling requires understanding not just *what* decision to make next, but *how* to organize exploration. SES verbalizes three components inside each trace: (1) **Sampling** (enumerate candidate actions in a canonical or reward-ordered sequence), (2) **Propagation** (predict / note resulting states or intermediate evaluations), and (3) **Backtracking** (Recover best path once goal or termination condition is detected, similar to Kazemi et al. (2023)). We refer to one supervised training example as a *Systematic Exploration Chain-of-Thought (SE-CoT)*.

SE-CoT linearizes a tree breadth-wise (default) with optional dead-end markers (e.g., `cut`) and concludes with a minimal backtrack listing. This consistent serialization induces an inductive bias for reusable loop structure ("expand frontier → annotate outcomes → proceed"). SE-CoT is still a CoT, but enriched with explicit branching structure that teaches systematic exploration of multiple solution paths.

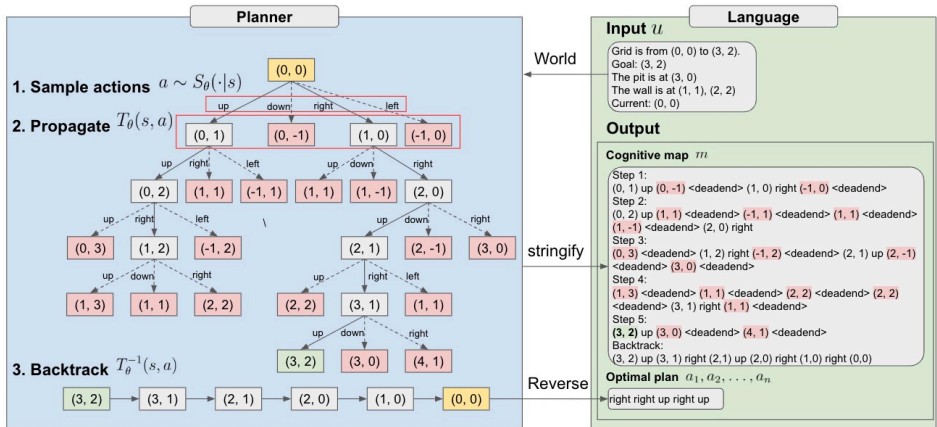

Figure 1: Systematic exploration supervision teaches models the complete algorithmic process as stringified Chain-of-Thought format, including (a) Sampling - systematic exploration of possible actions, (b) Propagation - predicting outcomes for each action, and (c) Backtracking - identifying the optimal path by reversing from the goal to the start.

## 3.2 METHOD 1: DIRECT ALGORITHMIC SUPERVISION

When complete search algorithms are available, we verbalize the entire search process:

**Algorithm 1** Direct Algorithmic Supervision

**Input:** Problem $p$ with known search algorithm, starting state $start$, goal state $goal$
**Output:** Systematic exploration trace $\tau$
Initialize: map $= []$, queue $= [start]$
**while** $goal \notin$ map **do**
   $s = $ POP(queue)
   **for** $a \in S(s)$ **do**
     {Sampling}
     $child = T(s, a)$ {Propagation}
     **if** $child \notin$ DEADEND **then**
       map$+ = (child, a)$
       queue$+ = child$
     **end if**
   **end for**
**end while**
$path = $ BACKTRACK$(goal, start, $map$)$
$\tau = $ VERBALIZE$($map$, path)$
**return** $\tau$

Each loop iteration is verbalized with (state, candidate actions, outcome / pruning rationale). Dead-end or dominated states are explicitly labeled, reducing ambiguity during learning. This differs from linear CoT which only supervises the single realized path: SES provides counterfactual siblings, teaching *why non-chosen actions were rejected*. See Appendix B for complete formatting templates.

### 3.3 METHOD 2: BOOTSTRAPPED SEARCH DISCOVERY

For open domains without explicit algorithms, we develop a bootstrapping technique that constructs synthetic search trees including both correct reasoning paths and plausible alternatives:

**Algorithm 2** Bootstrapped Search Discovery

**Input:** Problem $p$, LM $\pi_\theta$, branching factor $k$, depth limit $d$
**Output:** Systematic exploration trace $\tau$
Initialize: frontier $= [\texttt{initial\_state}(p)]$, explored $= []$, tree $= []$
gold_path $= $ GETGOLDSOLUTION$(p)$
**while** frontier $\neq \emptyset$ AND $|$explored$| < d$ **do**
   current $= $ frontier.$pop(0)$
   actions $= []$
   **for** $i = 1$ to $k$ **do**
     **if** current $\in$ gold_path AND $i = 1$ **then**
       $a = $ GETGOLDACTION(current, gold_path)
     **else**
       $a \sim \pi_\theta($current, context$)$ {Sample plausible alternative}
     **end if**
     next_state $= $ EXECUTE$(a, $current$)$
     actions.$append((a, $next_state$))$
     **if** next_state is valid AND next_state $\notin$ explored **then**
       frontier.$append($next_state$)$
     **end if**
   **end for**
   tree.$append(($current, actions$))$
   explored.$append($current$)$
**end while**
$\tau = $ VERBALIZE$($tree, gold_path$)$
**return** $\tau$

**Guaranteeing a valid solution.** We inject the gold action (when known) as the first branch at each gold state so every constructed tree contains at least one valid solution path. Remaining siblings are sampled from $\pi_\theta$ to approximate plausible alternatives.

**Reward-based ordering.** After expansion we sort the frontier by a lightweight reward heuristic $R$ (Inspired by RAP(Hao et al., 2023)) to prioritize seemingly promising nodes. This ordering supervision trains the model to *rank* expansions, not merely list them.

**Deduplication.** We filter candidate children that semantically duplicate an existing sibling (e.g., identical normalized arithmetic expression or logically equivalent intermediate statement). This encourages diversity and signals that repetition is uninformative, nudging the model toward novel branch generation.

## 4 TASK SETTINGS AND EXPERIMENTAL SETUP

We evaluate systematic exploration supervision across both controlled and open-ended reasoning tasks. Our experimental design allows systematic assessment of scaling capabilities while controlling for confounding factors.

### 4.1 TASK DOMAINS

**Gridworld.** We use textualized Gridworld (Brown, 2015) as our primary controlled evaluation environment. States are textual grid descriptions with agent position, goal, obstacles (walls), and pits. Actions are {up, down, left, right} with boundary and obstacle collision handling. Episodes terminate upon reaching the goal or exceeding a step limit (set to 2×grid perimeter). Each environment contains exactly one valid path from start to goal, ensuring deterministic optimal solutions.

Training environments range from 2×2 to 10×10 grids, while test environments scale up to 20×20. Detailed construction procedures, complexity analysis, and coordinate placement strategies are provided in Appendix C.

**Game of 24.** A combinatorial puzzle requiring arithmetic operations on four integers (1–9) to reach exactly 24 (Yao et al., 2023).[1] States represent multisets of available numbers/expressions; actions apply binary operators $\{+, -, \times, \div\}$ to any pair. Episodes terminate when a single number remains (success if equals 24, failure otherwise). Training: 1,200 instances; testing: 100 instances.

**GSM8K.** Grade-school math word problems requiring multi-step arithmetic reasoning (Cobbe et al., 2021). We decompose solutions into intermediate expressions where each state is a reasoning trace while the corresponding action is the next sentence. We use the RAP methodology (Hao et al., 2023) to provide reward signals for branch evaluation during bootstrapping.

**ProntoQA.** Logical reasoning tasks requiring multi-step deductive inference (Saparov & He, 2023). States are sets of known facts; actions instantiate inference rules (e.g., transitivity, contraposition). We use the RAP methodology (Hao et al., 2023) to provide reward signals for branch evaluation during bootstrapping. Deduplication employs canonical forms of predicate chains to avoid redundant explorations.

### 4.2 EXPERIMENTAL CONFIGURATION

We utilize Llama-3-8B model (Touvron et al., 2023) as a base (unless specified) with a maximum token length of 8,192 per instance. We train the model for 1 epoch for each task. We use one 8 Nvidia A100 node for both training and inference. For the training steps, we use FSDP framework (Zhao et al., 2023) and cosine annealing learning rate scheduler (Loshchilov & Hutter, 2017) for 1 epoch. We utilize bfloat16 floating-point format and a warmup ratio of 0.03. We set the weight decay as 0.

---

[1]While the Game of 24 has systematic algorithms for optimal solving, we use our bootstrapped approach (Algorithm 2) because verbalizing complete systematic search would result in prohibitively long context lengths, hence we classify it as an open-domain result.

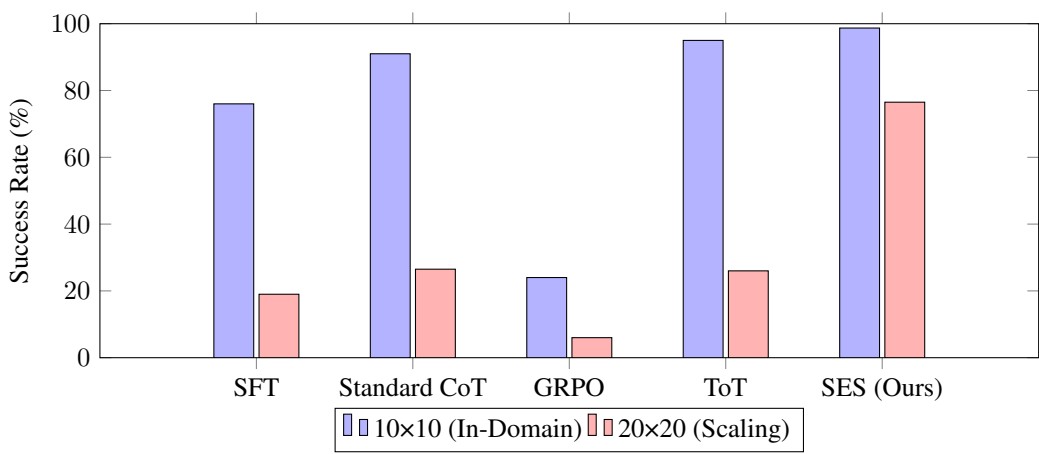

Figure 2: Success rates showing scaling performance. Systematic exploration supervision maintains substantially higher performance (76.5%) when scaling to larger environments compared to other approaches.

We set the batch size of 2 for each GPUs, so the effective batch size is 16 per step. For inference, we use VLLM framework (Kwon et al., 2023).

## 5 STRUCTURE SUPERVISION IN GRIDWORLD

We evaluate systematic exploration supervision in textualized Gridworld, where we can systematically control complexity and measure extrapolation capabilities.

### 5.1 EXPERIMENTAL SETUP AND ENVIRONMENT CONFIGURATION

**Gridworld Construction.** Training environments use grid sizes 2×2 to 10×10 with 15-25% obstacle density, single goal per episode, and 5 random seeds per size configuration. Test environments extend to sizes up to 20×20 with similar obstacle density. To minimize coordinate bias during extrapolation testing, grid placement uses uniform sampling within (0,0) to (19,19) bounds, ensuring the entire grid fits within this range while varying the starting position.

**Baseline Methods.** We compare against: (1) **SFT**: Standard supervised fine-tuning on direct action sequences, (2) **Standard CoT**: Chain-of-thought reasoning with step-by-step action justification, (3) **GRPO**: Group relative policy optimization using environmental rewards (Shao et al., 2024), (4) **Tree-of-Thought (ToT)**: Inference-time exploration with external search (Yao et al., 2023), and (5) **Systematic Exploration**: Our approach with complete search process supervision.

**Experimental Variants.** We evaluate several ablation conditions: **w.o. Systematic Search** removes multi-branch exploration; **w.o. Backtrack** omits the path extraction component; **Mark/Unmark** variants indicate whether dead-end states are explicitly labeled; **Forward/Backward Construction** refers to the direction of trace generation (start-to-goal vs goal-to-start).

### 5.2 MAIN RESULTS: SCALING PERFORMANCE

Figure 2 demonstrates our key finding: systematic exploration supervision enables effective scaling while other approaches fail dramatically. Our approach achieves a remarkable 4× improvement over standard supervised fine-tuning when extrapolating to larger environments (76.5% vs 19.0%). Notably, SFT suffers a catastrophic 57 percentage point drop when scaling beyond training complexity, while our method maintains robust performance with only a 22 point drop. GRPO exhibits particularly poor cold-start behavior, degrading from already-low 24% to merely 6% in scaling scenarios. Most surprisingly, Tree-of-Thought, despite achieving 95% success in-domain, fails to maintain its inference-time search advantage when complexity scales (dropping to 26%), suggesting

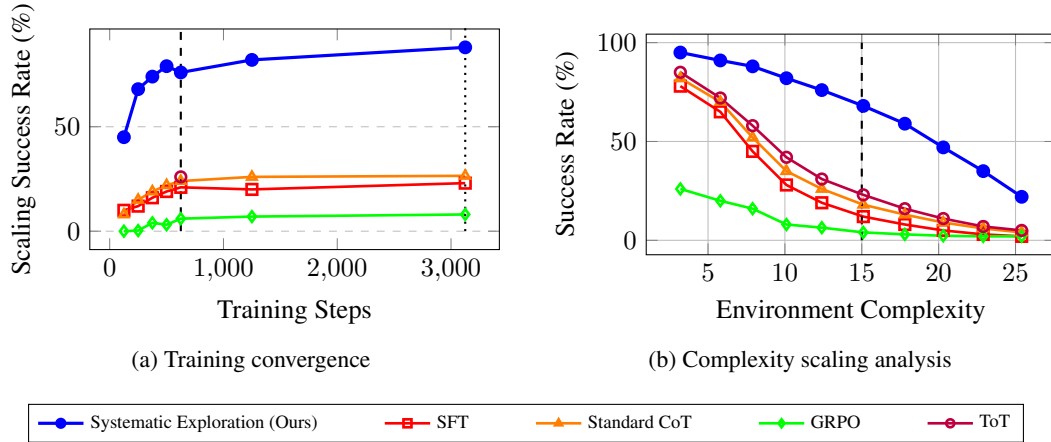

(a) Training convergence  (b) Complexity scaling analysis

Figure 3: Learning dynamics and complexity scaling. (a) Training convergence shows systematic exploration supervision reaches 79% scaling performance by step 500, while baselines plateau below 30%. The dashed line at step 625 marks our checkpoint; dotted line shows 5× extended training to observe baseline convergence. (b) Success rate vs environment complexity demonstrates graceful degradation for our approach vs. cliff-like failure for baselines.

that external search mechanisms may not transfer effectively across complexity boundaries without proper internalized search procedures.

We also conduct ablation studies of how each key factor of systematic search contributes to the improvements. Aside from SE-CoT itself, backtracking adds moderate benefits (+15 points), while dead-end marking shows mixed effects. Backward construction consistently outperforms forward construction, aligning with LAMBADA findings (Kazemi et al., 2023). Detailed ablation analysis across multiple construction variants is provided in Appendix D.

## 5.3 CONVERGENCE ANALYSIS

Figure 3 demonstrates key findings across training dynamics and complexity scaling. Our approach achieves rapid convergence (79% scaling performance by step 500, representing 16% of total training) and demonstrates superior learning efficiency with the first 250 training steps alone exceeding final baseline performance. We report results at step 625 (our checkpoint), but trained baselines 5× longer to observe their convergence behavior. Unlike baselines that plateau below 30% regardless of extended training, our method continues improving throughout training, reaching 88% at convergence. This sustained improvement suggests systematic exploration supervision provides qualitatively different learning signals that teach *algorithmic reasoning patterns* rather than memorizing (linear) solutions. The complexity analysis reveals graceful degradation across the entire spectrum, maintaining 22% success at complexity level 25 (67% beyond training boundary) while SFT collapses to 2%. This smooth scaling pattern strongly suggests our approach learns transferable problem-solving procedures rather than environment-specific heuristics.

## 5.4 PARAMETER SCALING AND FEW-SHOT ANALYSIS

**Parameter Efficiency Analysis.** Our experiments across the Qwen 2.5 model family (0.5B to 7B parameters) (Qwen et al., 2025) reveal striking parameter efficiency characteristics. Most remarkably, our 0.5B model achieves 68% scaling performance—dramatically outperforming 7B models trained with alternative approaches (26% for Standard CoT, 26% for ToT, 19% for SFT, 6% for GRPO). This represents a 14× parameter efficiency advantage, where our smallest model with 0.5B parameters outperforms baselines that use 14× more parameters.

The scaling curve for our method is notably flat (68% → 75% from 0.5B to 7B), suggesting that systematic exploration supervision captures *algorithmic competencies* that are largely independent of parameter count. In contrast, baseline methods show steeper parameter dependence: Standard

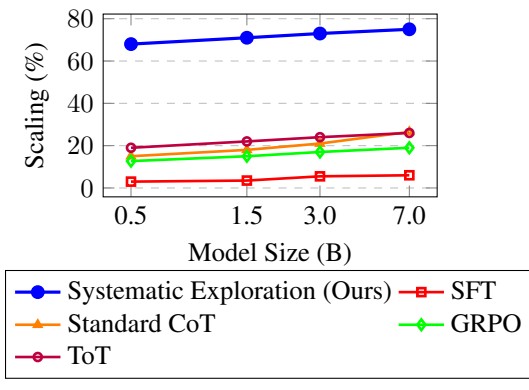

(a) Parameter scaling

| Model | 0-shot | 1-shot | 2-shot | 4-shot |
|---|---|---|---|---|
| *Standard CoT* | | | | |
| Llama-3 70B | 8% | 15% | 22% | 28% |
| GPT-4o | 12% | 19% | 26% | 31% |
| *SE-CoT* | | | | |
| Llama-3 70B | 0% | 0% | 0% | 0% |
| GPT-4o | 0% | 0% | 0% | 0% |
| GPT-o1 | 0% | 0% | 0% | 10% |

(b) Few-shot prompting results

Figure 4: Parameter efficiency and prompting analysis. (a) Our 0.5B model (68%) dramatically outperforms 7B baselines, showing 14× parameter efficiency. (b) SE-CoT cannot be elicited through prompting, requiring explicit training.

CoT improves from 15% to 26.5% (77% relative improvement), while SFT shows modest gains from 12.7% to 19% (50% relative improvement). This fundamental difference indicates that our approach teaches structured reasoning procedures that generalize across model scales, rather than relying on the brute-force memorization capacity that typically correlates with parameter count.

**Few-Shot Prompting Impossibility.**   The few-shot analysis reveals a fundamental cognitive distinction between systematic exploration supervision and conventional reasoning approaches. While standard Chain-of-Thought can be readily induced through few-shot prompting (with strong models achieving 28-31% performance), SE-CoT traces remain completely inaccessible through prompting alone across all tested conditions. Even GPT-o1, despite its sophisticated reasoning capabilities and extensive training, achieves only 10% success in the 4-shot condition, which is statistically indistinguishable from random performance.

## 6  PSEUDO-STRUCTURE SUPERVISION IN OPEN-ENDED DOMAINS

We next study how structure supervision helps in open-world domains by applying our bootstrapping technique (Method 2) to construct synthetic search trees for tasks without explicit algorithms.

### 6.1  OPEN-DOMAIN RESULTS

Figure 5 demonstrates the versatility of our bootstrapping technique across three distinct open-domain reasoning tasks, each representing different computational challenges:

**Game of 24: Large Search Space Validation.**   This combinatorial puzzle exemplifies scenarios where systematic exploration supervision excels. Our approach achieves 47% success—a remarkable 30 percentage point improvement over the next-best method (ToT at 17%) and a 2350% improvement over zero-shot performance. The dramatic performance gain in this discrete optimization task validates our hypothesis that structured exploration particularly benefits problems with large branching factors and sparse solution distributions. The fact that even sophisticated inference-time methods like ToT struggle (17%) while our offline supervision succeeds suggests that *internalized* search procedures can be more effective than external search when facing combinatorial complexity.

**GSM8K: Competent Baseline Analysis.**   Grade-school math problems reveal important limitations of our approach. GRPO achieves the highest performance (85%) while our method reaches 82.4%—a competitive but not superior result. Critically, the strong baseline performance (81.9% SFT vs 59.8% zero-shot) indicates that standard supervision already captures much of the required arithmetic reasoning patterns. This suggests that systematic exploration supervision provides maximal benefits

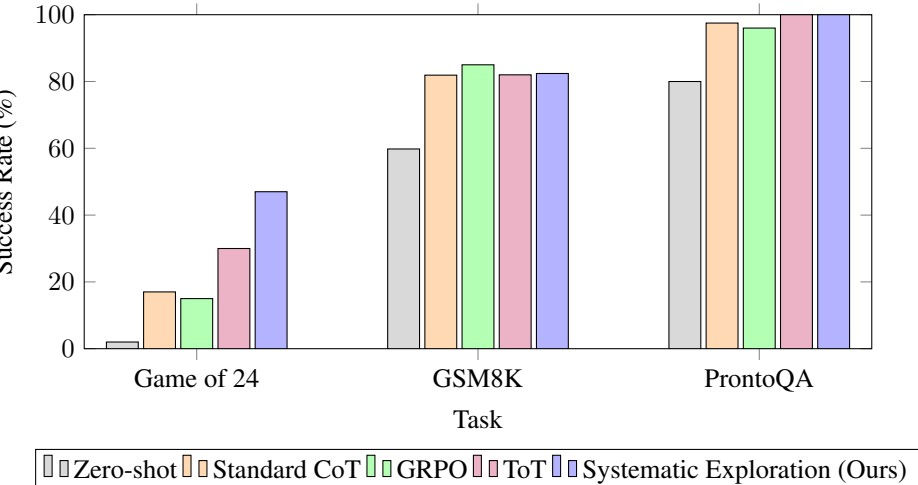

Figure 5: Performance across open-domain reasoning tasks. Systematic exploration supervision shows substantial improvements on Game of 24 (47% vs 0% zero-shot), competitive performance on GSM8K (82.4% vs 85% GRPO), and strong results on ProntoQA (100% matching ToT). Zero-shot baselines establish task difficulty.

when base competence is low, with diminishing returns when conventional approaches already achieve high performance.

**ProntoQA: Structured Logic Ceiling Effects.** Both our method and ToT achieve perfect performance (100%), substantially outperforming the 80% zero-shot baseline. This ceiling effect reveals that systematic exploration and inference-time search are both highly effective for structured logical deduction tasks with clear rule-based reasoning patterns. The convergence to perfect performance suggests that both approaches successfully capture the required deductive reasoning procedures.

**Task Characterization.** These results reveal clear patterns in method effectiveness. Systematic exploration supervision excels when: (1) base model performance is poor (0-20%), (2) large search spaces require systematic exploration, (3) problems involve scaling beyond training complexity, and (4) algorithmic structure exists or can be constructed. Alternative approaches are preferred when base models perform well (>50%), well-defined reward functions enable RL optimization, or domain knowledge dominates systematic search requirements.

## 7 CONCLUSION & FUTURE WORKS

We introduce systematic exploration supervision, a process supervision framework that teaches complete search processes to enable scaling beyond training complexity. Our approach addresses cold start problems in complex reasoning by providing structured supervision of exploration strategies rather than just correct answers. Key findings include substantial scaling improvements (76.5% vs 19%), "unpromptable" nature requiring explicit training, and method effectiveness depending on task characteristics. The framework offers a principled approach to teaching systematic problem-solving strategies with clear guidance on when this approach is preferable to alternatives.

While SES shows possibilities in our selected domains, several limitations exists. First, SE-CoT exponentially increases output tokens which may reduce effective batch size and approach context window limits (we chose 20×20 to remain inside 8,192 tokens). Also the comparable results on GSM8K suggest that our approach is not as effective when the base model is already competent. Future work might include how to improve the performance when the base model is already competent, how to improve the performance on general tasks, and how to improve the performance on tasks with high variance in branching factor.

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

# A    RELATED WORK

Our work intersects with several key research areas:

**Process Supervision and Chain-of-Thought.**    Process supervision (Uesato et al., 2022; Lightman et al., 2023) provides intermediate guidance during reasoning by supervising step-by-step solutions rather than just final answers. Recent work (Wang et al., 2024a) demonstrates that process-level supervision consistently outperforms outcome supervision across mathematical reasoning tasks. Chain-of-Thought (CoT) reasoning (Wei et al., 2023) teaches sequential reasoning through step-by-step demonstrations, showing improvements in arithmetic, logical reasoning, and planning tasks. However, traditional CoT approaches focus on single reasoning paths and fail to capture the systematic exploration of alternative solution strategies that characterizes human problem-solving.

**Reasoning via Supervised Chain of Thought (CoT) Learning**    Recent advances have shown that language models can learn to reason by imitating step-by-step reasoning processes (Wei et al., 2023; Kojima et al., 2023; Wang & Zhou, 2024). While standard CoT approaches focus on providing intermediate reasoning steps to reach a solution, research has expanded to include fine-tuning on CoT examples (Kim et al., 2023) and distilling reasoning into models without explicit verbalization (Deng et al., 2023; 2024). Most closely related to our work, Yang et al. (2022) proposed Chain of Thought Imitation, though their approach focuses on imitation learning of sequential procedures rather than the complete algorithmic planning process we explore. Despite these advances, conventional CoT methods typically capture a single reasoning path to a known solution rather than the full algorithmic process of exploration and decision-making, limiting their extrapolation capabilities.

**Reasoning via Reinforcement Learning**    The current state-of-the-art in reasoning capabilities comes from models trained with reinforcement learning techniques. Chu et al. (2025) established the influential paradigm that "SFT memorizes, RL generalizes," pushing the field toward extensive RL for reasoning enhancement. Approaches like RLHF (Ouyang et al., 2022) and more recently GRPO (Shao et al., 2024) have produced models with enhanced reasoning abilities, exemplified by DeepSeek-R1 (DeepSeek-AI et al., 2025), Phi-4 (Abdin et al., 2025), and other commercial systems. Alternative RL-based methods include Direct Preference Optimization (Rafailov et al., 2023) and Reinforced Self-Training (Gulcehre et al., 2023). While effective, these approaches face significant challenges including reward design complexity, computational demands, and the chicken-and-egg problem we identified earlier.

**Planning in Language Models**    Various approaches have explored enhancing language models' planning capabilities. Tree-structured exploration methods like Tree-of-Thought (Yao et al., 2023) and more theoretically grounded analyses of planning capability (Wang et al., 2024b) have shown promise for complex reasoning. Systems like ReAct (Yao et al., 2022), SwiftSage (Lin et al., 2024), and Reflexion (Shinn et al., 2023) combine reasoning with action in interactive environments. Merrill & Sabharwal (2024) and Pfau et al. (2024) provide theoretical insights into how transformers perform planning computations. Particularly relevant to our approach, Hao et al. (2023) frame reasoning as planning with an internal world model, though without explicitly teaching the planning algorithm through supervised learning as we propose. In spatial domains specifically, recent evaluations have revealed limitations in LLMs' spatial reasoning capabilities (Xu et al., 2024; Li et al., 2024; Yamada et al., 2024), highlighting the challenges that our approach addresses.

**Transformer Limitations and Extrapolation**    Our work also connects to fundamental research on transformer architecture limitations. Studies by Merrill & Sabharwal (2023), Dziri et al. (2024), and Peng et al. (2024) identify inherent constraints in transformers' ability to perform certain algorithmic computations, particularly in tasks requiring compositional generalization. Balestriero et al. (2021) argues that learning in high dimensions fundamentally involves extrapolation rather than interpolation, providing theoretical grounding for why extrapolation is such a challenging capability for neural networks to acquire.

**Spatial Reasoning in Language Models.**    Recent studies evaluate language models' spatial understanding and cognitive mapping capabilities (Xu et al., 2024; Momennejad et al., 2023; Yamada et al., 2024). These works reveal that while language models show some spatial reasoning abilities,

they struggle with forming cognitive maps necessary for effective planning in complex environments. Our work addresses this limitation by explicitly training models to construct and utilize systematic exploration strategies.

# B  INPUT-OUTPUT FORMAT

## B.1  FORMAT EXAMPLES: GRIDWORLD

---

**Gridworld Input Instance:**
**USER:** You are given a rectangular gridworld, where you can move up, down, left, or right as long as each of your x, y coordinate is within 0 to the x, y size of the grid. If you move up, your y coordinate increases by 1. If you move down, your y coordinate decreases by 1. If you move left, your x coordinate decreases by 1. If you move right, your x coordinate increases by 1.
You will interact with the girdworld environment to reach the goal state, while avoiding the pit and the wall. You cannot move through the wall or move outside the grid. If you fall into the pit, you lose. If you reach the goal, you win. For each of your turn, you will be given the possible moves.
You should respond your move with either one of 'up', 'down', 'left', or 'right'.
**ASSISTANT:** OK
**USER:** Grid is from (7, 7) to (18, 13). Goal: (18, 13) Current: (7, 7) The pit is at (16, 9), (17, 7), (14, 8), (11, 9), and (16, 11). The wall is at (11, 7), (15, 10), (18, 12), (9, 9), (8, 7), (17, 10), (9, 12), (12, 7), (10, 13), (15, 12), (17, 9), (9, 11), (17, 12), (8, 9), (13, 11), (10, 9), (12, 9), (11, 11), (12, 12), (16, 10), (15, 8), (9, 7), (8, 11), (7, 13), (13, 13), and (13, 10).

---

**Gridworld Output Instance (Direct output):**
**ASSISTANT:** up up up up

---

**Gridworld Output Instance (Standard CoT):**
**ASSISTANT:** <think>
Step 1
up
(7, 8)[SEP1][SEP3]
Step 2
up
(7, 9)[SEP1][SEP3]
Step 3
up
(7, 10)[SEP1][SEP3]
Step 4
up
(7, 11)[SEP1][SEP3]
<\think>
<answer>
up up up up
<\answer>

---

**Gridworld Output Instance (SE-CoT):**
**ASSISTANT:** <think>
Step 1
up
(7, 8)[SEP1]
down
(7, 6)[SEP2]
left
(6, 7)[SEP2]
right
(8, 7)[SEP2][SEP3]
Step 2

---

up
(7, 9)[SEP1]
down
(7, 7)[SEP2]
left
(6, 8)[SEP2]
right
(8, 8)[SEP1][SEP3]
<\think>
<answer>
up up up up
<\answer>

## B.2    FORMAT EXAMPLES: GAME OF 24, PRONTOQA, GSM8K

### B.2.1    FORMAT EXAMPLES: GAME OF 24

**Game of 24 Input Instance:**
**USER:** Input: 2, 2, 6, 8

**Game of 24 Output Instance (Standard CoT):**
**ASSISTANT:** <think>
Step 1
8 - 2 = 6[SEP1]
Step 2
6 - 2 = 4[SEP1]
Step 3
6 * 4 = 4[SEP1][SEP3]
<\think>
<answer>
(8 - 2) * (6 - 2) = 24
<\answer>

**Game of 24 Output Instance (SE-CoT):**
**ASSISTANT:**<think>
Step 1
2 + 2 = 4[SEP1]
2 * 2 = 4[SEP1]
8 - 2 = 6[SEP1]
8 * 6 = 48[SEP1]
8 / 6 = 4/3[SEP1][SEP3]
Step 2
4 + 8 = 12[SEP1]
...
<\think>
<answer>
(8 - 2) * (6 - 2) = 24
<\answer>

### B.2.2    FORMAT EXAMPLES: PRONTOQA

**ProntoQA Input Instance:**
**USER:** Arthropods are small.  Insects are not eight-legged.  Lepidopterans are insects. Invertebrates are animals.  Every spider is eight-legged.  Insects are arthropods.  Every butterfly is a lepidopteran. Animals are multicellular. Arthropods are invertebrates. Sam is a butterfly.

True or false: Sam is not eight-legged.

**ProntoQA Output Instance (Standard CoT):**
**ASSISTANT:**<think>
Step 1
Sam is a butterfly.[SEP1][SEP3]
Step 2
Every butterfly is a lepidopteran.[SEP1][SEP3]
Step 3
Sam is a lepidopteran.[SEP1][SEP3]
Step 4
Lepidopterans are insects.[SEP1][SEP3]
Step 5
Sam is an insect.[SEP1][SEP3]
Step 6
Insects are not eight-legged.[SEP1][SEP3]
Step 7
Sam is not eight-legged.[SEP1][SEP3]
Step 8
The answer is true.[SEP4][SEP3]
<\think>
<answer>
true
<\answer>

**ProntoQA Output Instance (SE-CoT):**
**ASSISTANT:** <think>
Step 1
Sam is a butterfly.[SEP1][SEP3]
Step 2
Every butterfly is a lepidopteran.[SEP1][SEP3]
Step 3
Sam is a lepidopteran.[SEP1][SEP3]
Step 4
Lepidopterans are insects.[SEP1][SEP3]
Step 5
Sam is an insect.[SEP1][SEP3]
Step 6
Insects are not eight-legged.[SEP1]
Insects are arthropods.[SEP1][SEP3]
Step 7
Sam is not eight-legged.[SEP1][SEP3]
Sam is an arthropod.[SEP1][SEP3]
<\think>
<answer>
true
<\answer>

### B.2.3 FORMAT EXAMPLES: GSM8K

**GSM8K Input Instance:**
**USER:** A robe takes 2 bolts of blue fiber and half that much white fiber. How many bolts in total does it take?

**GSM8K Output Instance(Standard CoT):**
**ASSISTANT:** <think>
Step 1
It takes $2 * 0.5 = << 2 * 0.5 = 1 >> 1$ bolt of white fiber.[SEP1][SEP3]
Step 2

So it takes $2 + 1 = << 2 + 1 = 3 >> 3$ bolts in total.[SEP1][SEP3]
Step 3
The answer is 3.[SEP1][SEP3]
<\think>
<answer>
3
<\answer>

**GSM8K Output Instance(SE-CoT):**
**ASSISTANT:** <think>
Step 1
It takes $2 * 0.5 = << 2 * 0.5 = 1 >> 1$ bolt of white fiber.[SEP1]
The robe takes 2 bolts of blue fiber.[SEP1][SEP3]
Step 2
So it takes $2 + 1 = << 2 + 1 = 3 >> 3$ bolts in total.[SEP1][SEP3]
It also takes half as much white fiber, which means it takes 1 bolt of white fiber (since half of 2 is 1).[SEP1][SEP3]
Step 3
The answer is 3.[SEP4][SEP3]
<\think>
<answer>
3
<\answer>

## C  GRIDWORLD CONSTRUCTION

While exploring the pure planning ability of the language model, we did not want the model to refuse to explore extrapolated data only because it has never seen the coordinate. To handle the bias, we adjust the starting position of the grid using uniform sampling while ensuring that the entire grid, with its x and y coordinates, fits within the range of 0 to 19. This method, illustrated in Figure 6, minimizes bias related to the unseen x and y coordinates by randomizing the starting point within the defined bounds.

### C.1  ENVIRONMENT CONFIGURATION

Training environments use grid sizes 2×2 to 10×10 with 15-25% obstacle density, single goal per episode, and 5 random seeds per size configuration. Test environments extend to sizes up to 20×20 with similar obstacle density. Grid placement uses uniform sampling within (0,0) to (19,19) bounds to minimize coordinate bias during extrapolation testing.

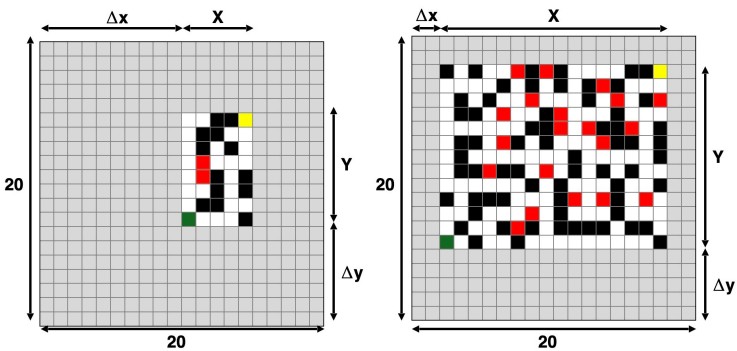

Figure 6: Visualization of configuring Gridworld instance for train(left) and test(right) dataset. To evaluate the extrapolation ability, we set the size of the grid as $X, Y \sim Unif(2, 10)$ for train and $X, Y \sim Unif(2, 20)$ for test. Also we set the starting point of the grid as $\Delta x \sim Unif(0, 20 - X), \Delta y \sim Unif(0, 20 - Y)$ for both train and test.

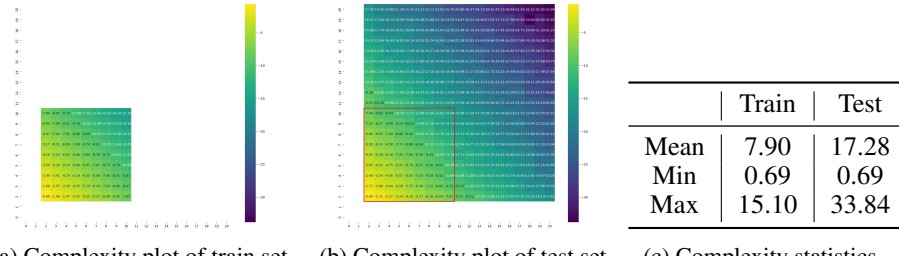

| | Train | Test |
|---|---|---|
| Mean | 7.90 | 17.28 |
| Min | 0.69 | 0.69 |
| Max | 15.10 | 33.84 |

(a) Complexity plot of train set    (b) Complexity plot of test set    (c) Complexity statistics

Figure 7: Complexity analysis of Gridworld environments. Each heatmap (a, b) shows the average complexity (Equation 1) of Gridworlds of size $x \times y$, where the darkness at each $(x, y)$ coordinate represents the mean complexity. **(a)** depicts the training set, and **(b)** shows the test set, with the red box (from $(2, 2)$ to $(10, 10)$) indicating the training boundary. **(c)** provides summary statistics of complexity for both train and test sets. Details on train/test dataset statistics can be found in Appendix C.2.

## C.2    INPUT/OUTPUT STATISTICS

Figures 8 and 9 present detailed statistics for input and plan token lengths, respectively. Similar to the complexity analysis discussed in Section 2, notable discrepancies exist between the train and test datasets, emphasizing the extrapolation challenges posed by larger and more complex Gridworld environments.

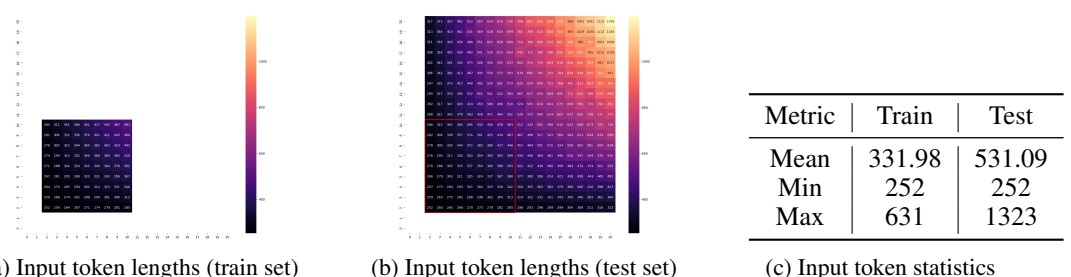

| Metric | Train | Test |
|---|---|---|
| Mean | 331.98 | 531.09 |
| Min | 252 | 252 |
| Max | 631 | 1323 |

(a) Input token lengths (train set)    (b) Input token lengths (test set)    (c) Input token statistics

Figure 8: Input length analysis of Gridworld environments. Heatmaps (a, b) represent the average input token length for Gridworlds of size $x \times y$, where the darkness at each $(x, y)$ coordinate indicates the mean token length. **(a)** shows the training set, and **(b)** shows the test set. The red box (from $(2, 2)$ to $(10, 10)$) outlines the training boundary. Summary statistics are provided in (c).

## D    COMPLETE ABLATION RESULTS

We conduct comprehensive ablation studies to identify the critical components of systematic exploration supervision. Table 5 shows the full ablation studies among different configurations with detailed analysis of each component:

**Systematic Search vs. Baseline Methods.**    The most significant performance gain comes from systematic search supervision itself. Removing systematic exploration ("w.o. All Components") reduces performance to 27.7%, comparable to standard CoT approaches. This 47-point improvement demonstrates that explicit multi-branch exploration supervision is the primary driver of scaling success.

**Backtracking Component.**    Adding backtracking provides moderate but consistent improvements (+15 percentage points from 29.6% to 76.5%). Backtracking teaches the model to identify and verbalize the optimal path after exploration, reinforcing the connection between search and solution extraction.

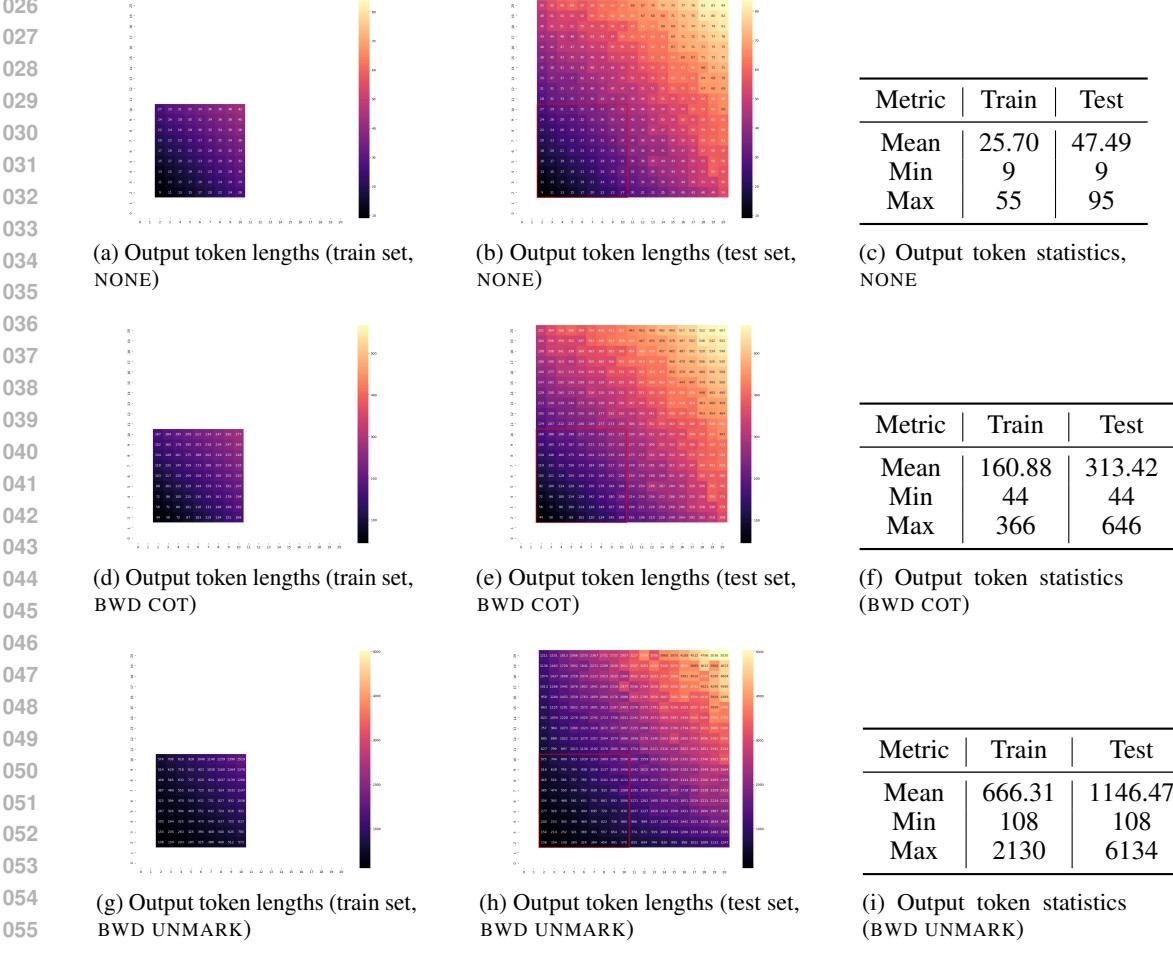

| Metric | Train | Test |
| --- | --- | --- |
| Mean | 25.70 | 47.49 |
| Min | 9 | 9 |
| Max | 55 | 95 |

(a) Output token lengths (train set, NONE)

(b) Output token lengths (test set, NONE)

(c) Output token statistics, NONE

| Metric | Train | Test |
| --- | --- | --- |
| Mean | 160.88 | 313.42 |
| Min | 44 | 44 |
| Max | 366 | 646 |

(d) Output token lengths (train set, BWD COT)

(e) Output token lengths (test set, BWD COT)

(f) Output token statistics (BWD COT)

| Metric | Train | Test |
| --- | --- | --- |
| Mean | 666.31 | 1146.47 |
| Min | 108 | 108 |
| Max | 2130 | 6134 |

(g) Output token lengths (train set, BWD UNMARK)

(h) Output token lengths (test set, BWD UNMARK)

(i) Output token statistics (BWD UNMARK)

Figure 9: Output length analysis of Gridworld environments. Heatmaps (a, b) represent the average plan token length for Gridworlds of size $x \times y$. The darkness at each $(x, y)$ coordinate indicates the mean token length. **(a, b)** depict the direct path approach, **(d, e)** show Chain of Thought (CoT) reasoning, and **(g, h)** illustrate cognitive map reasoning. The red box (from $(2, 2)$ to $(10, 10)$) outlines the training boundary. Summary statistics for each approach are provided in (c, f, i).

**Dead-end Marking.** Explicit marking of invalid states shows mixed effects: beneficial for some tasks but potentially constraining for others. The BWD UNMARK versions (76.5%) slightly outperform BWD MARK versions (70.5%) in optimal planning, suggesting that implicit learning of invalid states may be more robust than explicit annotation.

**Construction Direction.** Backward construction (goal-to-start) consistently outperforms forward construction across all variants, aligning with LAMBADA findings (Kazemi et al., 2023) that backward chaining enhances planning effectiveness.

**Terminology Definitions.** The following terminologies are used throughout the ablation analysis:

- BWD UNMARK: Backward construction without explicit dead-end marking
- BWD MARK: Backward construction with explicit dead-end marking
- FWD UNMARK: Forward construction without explicit dead-end marking
- FWD MARK: Forward construction with explicit dead-end marking
- W.O. BACKTRACK: Configuration without backtracking component
- W.O. ALL COMPONENTS: Configuration without systematic search or backtracking

Systematic search provides +45-47 percentage point improvements for optimal planning. All differences are statistically significant (p<0.001).

| | w.o. Systematic Search | | with Systematic Search | |
|---|---|---|---|---|
| Optimal Planning | w.o. Backtrack | w.o. All Components | Unmark Deadends | Mark Deadends |
| BWD Construction | 0.296 ± 0.021 | 0.277 ± 0.019 | **0.765 ± 0.024** | 0.705 ± 0.022 |
| FWD Construction | 0.295 ± 0.020 | 0.290 ± 0.023 | **0.618 ± 0.026** | 0.585 ± 0.025 |

Table 5: Complete ablation results with standard deviations across 5 seeds.

# E COMPREHENSIVE GRIDWORLD ANALYSIS

## E.1 DETAILED PERFORMANCE VISUALIZATION

| | SFT Baseline | Standard CoT | Systematic Exploration |
|---|---|---|---|
| Success Heatmap | 19.0% | 26.5% | 76.5% |

Table 6: Performance heatmaps across different grid sizes. Darkness indicates success rate at each (x,y) coordinate. Red boxes mark training boundary (10×10). Systematic exploration supervision maintains strong performance well beyond training bounds while baselines show sharp degradation.

Table 6 provides comprehensive visualization of performance across all grid sizes. We can observe that all methods perform well within the red box (10×10 training boundary), but both SFT and Standard CoT show rapid performance drops outside training bounds. On the other hands, our approach maintains significant performance even in 20×20 environments

## E.2 ABLATION STUDY VISUALIZATION

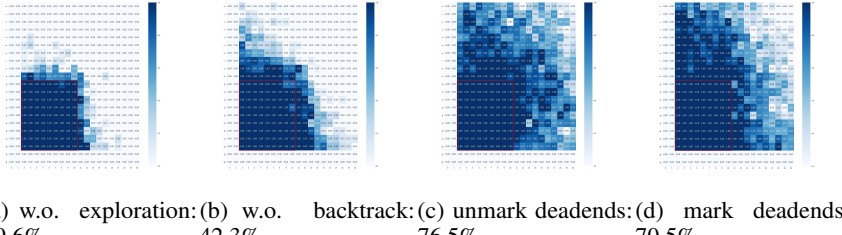

(a) w.o. exploration: 29.6%      (b) w.o. backtrack: 42.3%      (c) unmark deadends: 76.5%      (d) mark deadends: 70.5%

Figure 10: Ablation study heatmaps showing the effect of different components. The progression clearly shows systematic exploration as the primary driver (+47 percentage points) and backtracking as secondary (+15 points).

Figure 10 visualizes the ablation study results across the scaling space. The progression from (a) to (c) demonstrates how each component contributes to scaling performance, with systematic

exploration providing the dramatic improvement from 29.6% to 76.5%. Figure 3 demonstrates that systematic exploration supervision scales smoothly with complexity, while other methods show sharp performance cliffs. This suggests our approach learns generalizable problem-solving strategies rather than memorizing specific patterns.

## F    USE OF LARGE LANGUAGE MODELS

We acknowledge the use of large language models to assist in the preparation of this manuscript. Specifically:

**Writing assistance.**    Large language models were used to aid in polishing and refining the writing throughout the paper, including improving clarity, grammar, and expression of technical concepts.

**Related work discovery.**    Large language models were employed for retrieval and discovery tasks, particularly in identifying and organizing relevant related work and ensuring comprehensive coverage of the literature.

All technical contributions, experimental design, implementation, analysis, and conclusions presented in this work are the original work of the authors. The use of LLMs was limited to editorial assistance and literature search support, and did not influence the core scientific contributions or findings reported in this paper.

