# OpenReview forum: "Systematic Exploration Supervision Enables Scaling Beyond Training Complexity"
_ICLR.cc/2026/Conference — Submitted to ICLR 2026_

### Official Review · Reviewer_puAH · 2025-10-28

**Soundness:** 3
**Presentation:** 3
**Contribution:** 2
**Rating:** 4
**Confidence:** 2

**Summary:**

This work is motivated by the observations: 1) SFT lacks the ability of exploration, and it can not adapt the model to different/more complicated tasks (even with same solving strategy); and 2) RL converges slowly and is typically data-inefficient due to low initial success rates. To address this, the paper proposes Systematic Exploration Supervision (SES), a framework that introduces sampling, propagation, and backtracking into model training. Experimental results show that SES outperforms Tree-of-Thought and GRPO.

**Strengths:**

1. The paper proposes an online search algorithm for post-training LLMs, allowing the model not to strictly follow SFT paths but to explore intelligent problem-solving routes via search.

2. Experimental results provide evidence of the effectiveness of the proposed method.

**Weaknesses:**

1. The online search policy is not a novel topic and there are many prior works on it (eg., Monte Carlo Tree Search and etc). This work lacks a thorough literature review comparing SES to prior algorithms.
2. This method requires that enough successful trajectories have been collected. For the problems the authors tried to solve, the problem complexity is low and obtaining successful paths is easy. However, it may not adapt to realistic settings where problems are much more complex and successful trajectories are rare.
3. It seems that the proposed algorithm counts and stores all possible routes in memory, which is inefficient if the problem has wide search space and requires long steps to solve.
4. Algorithms are not clearly stated. For examples, the implementations of Backtrace, Verbalize, GetGoldAction in Algorithm1&2 are not explained.
5. Lack of experimental comparison to the other online search algorithms.
6. Lack of experimental comparison on more complex problems.

**Questions:**

1. In Algorithm 1, Line 168-175, It seems that all $a\in S(s)$ are added to the queue sequentially, resulting in a queue like [start,$a_1,a_2,...$] for $a_i$ in $S(s)$. What is the intuition behind this queue structure?
2. In the Gridworld experiments, were all possible routes explicitly explored and stored during training?
3. Line 183 mentions that “Deadend or dominated states are explicitly labeled.” Does SES only apply to problems where such states can be clearly defined?
4. What are the SFT results in Figure 5?

---

> ### Author Response · Authors · 2025-11-12
> **W1 & W5**
>
> Thank you for engaging deeply with our work. I think there might be a fundamental misunderstanding about what we're proposing, and I'd really like to clear this up because it affects how we should interpret your concerns.
>
> Let me start with what I think might be the core confusion. When you mention "online search policy" and comparisons to Monte Carlo Tree Search, I'm wondering if you're reading our method as another inference-time search algorithm like Tree-of-Thought? Because that's actually not what we're doing at all, and if that's the impression we gave, we need to fix our framing immediately.
>
> Here's the thing: **SES isn't a new way to search at test time, but rather a training method that teaches models to internalize the search process itself.** Think of it this way: ToT is like a student who doesn't understand the problem-solving method, so they frantically try 100 different approaches in parallel hoping one works by chance. SES is like actually teaching the student the systematic method so they can solve it reliably in one attempt. SES is like teaching them to do arithmetic in their head. At test time, our models just generate text autoregressively (one forward pass), but that text happens to contain systematic exploration because they learned the algorithm during training.
>
> This is why we only compared to ToT in our experiments, and I think this might address several of your concerns at once. ToT represents the upper bound of what parallel sampling methods can achieve. It's essentially doing pass@b^d, trying every possible path up to depth d with branching factor b. Test-time scaling methods like RAP and LATS (which use MCTS; along with non-MCTS methods like Self Consistency) are fundamentally just reordering those same samples based on rewards or value estimates. They're all trying to externally recover the search algorithm through brute-force parallel sampling.
>
> But here's the problem with all of them: **the samples are independent, not causally connected.** Each sampled path doesn't actually learn from the others during inference. They're running the same base model that doesn't know how to search, just hoping that by rolling the dice enough times, one path works out. That's why ToT still only gets 26% on our 20x20 grids even though it's doing expensive tree search, since the base model never learned to systematically explore.
>
> When our model generates "try going left... no that hits a wall... try going up instead...", that's not external search happening. That's the model having learned that this is how you solve navigation problems.
>
> Does this distinction make sense? I want to make sure we're aligned on what we're actually claiming before diving into the specific technical points.

---

> ### Author Response · Authors · 2025-11-13
> **W2 & W6**
>
> Meanwhile, let us also address your concern on problem complexity and requiring successful trajectories. You raise the point about problems where successful trajectories are rare and that we need experiments on "more complex problems," and we think there might be a mismatch in how we're defining “complexity” in general here.
>
> Look at the actual numbers: Game of 24 gets 2% zero-shot on frontier models. Our 20x20 Gridworld gets 0%, even with few-shot prompting (Figure 4b). These are not toy problems, but rather legitimately hard ones for even the most capable models available. The fact that they have clear structure and verifiable solutions doesn't make them easy; It makes them good benchmarks for controlled study.
>
> And this is exactly where Algorithm 2 comes in. We designed the bootstrapping procedure specifically for cases even when the base model can't reliably solve problems. The key trick is that we **guarantee at least one valid solution is in every training tree** (line 202-203), then fill the rest with plausible alternatives from the model’s own rollout. So even when base success is 2%, we can construct rich search trees that teach systematic exploration. That's why we get an improvement to 47% on Game of 24, compared to 17% with SFT from standard CoT.
>
> Now here's where I think there might be a misunderstanding about data requirements. You mention that our method requires successful trajectories and might not work when they're rare. Actually, we need **exactly the same level of data source as standard SFT**: one correct solution per training example. If you can train a model with SFT, you can train with SES. The only problems we can't handle are ones where you literally cannot obtain even a single verified solution (unsolved problems, unverifiable tasks), which is also where SFT fails.
>
> Moreover, our data requirements are actually **less stringent than the RL baselines we compare against**. GRPO needs thousands of environment rollouts per problem to learn from sparse rewards, which is why it collapses to 6% on our scaling task, because cold start is brutal when you need massive exploration to get any signal. We just need one verified solution plus the model's own samples for diversity. So the real comparison isn't "SES vs impossible problems with no solutions", rather "SES vs RL methods that need orders of magnitude more data and still fail when base competence is low."
>
> In conclusion, we are not claiming to build the best agent on the hardest possible benchmarks. We're proposing a novel training methodology and showing it works under controlled conditions where we can actually measure what's happening. That's scientific methodology where we need to control variables, measure carefully, and understand the mechanism. Game of 24 at 2% base accuracy and Gridworld that stumps GPT-o1 are already complex problems by any reasonable definition. The contribution isn't "look how well we do on fancy benchmark that just came out this year." The contribution is "here's a training methodology that shows a specific phenomenon: teaching complete search traces enables scaling beyond training complexity in a way that existing methods don't."
>
> If we had run on some massive real-world task, would you be more convinced? Or would your followup question be 'how do you isolate the effect of your training method from confounds?' Our choice of testbed was validated (Section 2) via the way that we could control complexity, measure it precisely (Equation 1 & Appendix C), and isolate the effect of our training method. We believe this is the right way to establish whether a new training methodology works. Nevertheless, we’re curious what experimental design would satisfy your concern about "complexity", if you think our proposition is further reinforced.

---

> ### Author Response · Authors · 2025-11-13
> **W3**
>
> You're right; We do verbalize extensive search traces, and this **does** increase memory and token usage substantially. But I want to reframe why this isn't actually a limitation of our method specifically, rather a fundamental characteristic of the reasoning paradigm we're exploring.
>
> Think about o1-like reasoning chains. The model generates long traces with many dead-ends and explorations that don't directly contribute to the final answer. You could argue that's "inefficient" because all the computation before the "aha moment" doesn't literally appear in the solution. But that misses the point: **the computation that induced the aha moment is precisely what enables robust reasoning.** The model needed to explore those paths, even the wrong ones, to develop confidence in the right one.
>
> Our approach is similar. We're verbalizing the full search process including branches that don't lead to the solution. A 10x10 grid produces about 666 output tokens versus ~25 for direct output (Appendix C, Figure 9). But this isn't waste, but it's helping the model to learn **how to think systematically** about navigation problems. The branches that hit walls or dead-ends are teaching moments, not inefficiency.
>
> Now, are there problems with very long reasoning chains? True. Models can hallucinate or lose coherence over extended traces. But that's a general challenge of the o1-like reasoning paradigm itself, not specific to SES. Our paper focuses on how to construct effective search-structured training data, not on solving all the engineering challenges of long-context reasoning. Advanced attention mechanisms and memory management techniques exist to handle long chains, just that we didn't focus on that aspect here.
>
> And crucially, for Algorithm 2, we're not actually exploring all k^d nodes naively. We deduplicate semantically equivalent nodes (line 208-209), which dramatically reduces the actual tree size. For concrete numbers: GSM8K with d=10, k=5, and temperature = 0.3 always fits comfortably within 8192 tokens. Game of 24 and ProntoQA with the same parameters (just different d) stay well within limits. We're not trying to cover the entire combinatorial space, but rather sampling enough diverse paths to teach the exploration procedure.
>
> So yes, our traces are longer than standard CoT. But if you accept that o1-style reasoning with its exploratory dead-ends is valuable, then you should accept that teaching models to generate such structured exploration is valuable too. The token overhead is the cost of teaching systematic thinking, not a bug in our approach.

---

> ### Author Response · Authors · 2025-11-13
> **W4 & Questions**
>
> For W4: This is a fair point. Sorry for missing it. We commented with the title "Clarification on algorithm implementation". We will make sure this is revised after this process.
>
> For Q1: It's implementing breadth-first search, and the intuition is that BFS gives us the shortest path (optimal solution) while also providing a rich training signal because we explore all states at depth d before going to depth d+1. This creates a consistent learning pattern of "expand all current options, evaluate them, proceed to next level."
>
> For Q2: Each Gridworld instance has exactly one valid path by construction (Appendix C explains this).
>
> For Q3: the "mark vs unmark" ablation in Figure 10 shows unmark actually works better (76.5% vs 70.5%), so explicit dead-end annotation isn't necessary. For open-domain tasks where there's no clear "dead-end," we just use reward signals (RAP style, Section 4.1.). If you can evaluate a reasoning step (even approximately), you can construct search trees.
>
> For Q4: "SFT" and "Standard CoT" are the same thing in Figure 5. For SFT on open-ended reasoning, we don't just give them answers but also the standard thinking chain as supervision.

---

### Official Review · Reviewer_iiWd · 2025-10-29

**Soundness:** 3
**Presentation:** 3
**Contribution:** 3
**Rating:** 8
**Confidence:** 3

**Summary:**

This paper introduces Systematic Exploration Supervision (SES), a supervision framework designed to improve the ability of LLMs to reason in structured environments such as Gridworld. Rather than training models to generate linear CoT, SES trains the model on systematically-generated multi-branch search trees that represent good exploration (backtracking, sampling alternative paths etc.) of the structured environment - these trees can be viewed as process-supervised samples that teach the model what good exploration "looks like".

During test time, the model is prompted to solve the task using linear CoT (SE-CoT). Crucially, the authors demonstrate that the SES-trained models improve significantly over RL, SFT, CoT-prompted and Tree-of-Thoughts baselines on tasks involving structured exploration even though search is conducted only with linear CoT. The authors argue that this arises from the fact that SES trains the model to internalise the search process. Furthermore, they show that this internalisation phenomenon allows the model to generalise to more complex environments than they were trained on (e.g. larger grids in Gridworld).

Through various ablation studies, the authors demonstrate the limits of SES: while SES significantly strengthens model performance on tasks where structured exploration is effective (e.g. Gridworld, Game of 24), SES provides less benefit on tasks where traditional linear search is effective (e.g. maths reasoning) and where the base model is already strong.

**Strengths:**

- The authors present an intuitive yet interesting method for teaching models to learn better search "skills" or "reasoning patterns" in structured environments. The results are strong and support their central claims.
- I appreciate the fact that they provide analysis on when their method does not work, which helps us better understand what SES actually teaches the model to do, and may help guide future research in this direction. These findings also help us distinguish between different kinds of reasoning tasks (e.g. structured exploration tasks like Gridworld vs. more linear, "depth-focussed" tasks like GSM8K), and highlights the fact that different strategies may be required to achieve large gains on these different tasks.
- The research is a timely contribution given the direction of the field - much research has been done scaling test-time compute (train models/create new inference-time algorithms such that generating more reasoning tokens at test-time -> better performance), whereas this paper demonstrates that using compute to instead generate better training data nonetheless remains a promising path.

**Weaknesses:**

I do not believe there are any fundamental flaws or weaknesses in the paper that make it unsound. I nonetheless have some thoughts on how it may be improved:
- It is unclear to me why and how performance degrades as we scale up the grid sizes - any search skills learned for small grids are equally applicable for larger grids (the model merely has to apply these skills more often). Clearly the internal search skill learned is not perfect - it would be interesting to see some analysis on the failure modes of the model to better understand the method's limitations.
- It could be beneficial to provide some examples of the training data (i.e. the exact inputs/labels you do SFT on).
- One potential criticism of this work is that Gridworld is a very simple task, and that increasing the size of the grid doesn't represent a meaningful increase in its difficulty, given that the same search operators are effective regardless of the grid size (and likewise for Game of 24), and that SES main still fall behind using test-time algorithms on more complex structured search tasks where a larger degree of skill/knowledge generalisation may be required (e.g. MiniHack https://arxiv.org/abs/2109.13202, Crafter https://github.com/danijar/crafter). As such, I think it would be useful to see SES applied to more difficult structured exploration tasks.
- One of the baselines is the SFT baseline, where the model is SFT'd on (successful?) trajectories. How many samples are used here vs. for the SES model? It would be interesting to compare the two in a compute-controlled setting (since each SES sample is much longer than a direct SFT trajectory, the compute-controlled comparison would require training on many more direct trajectories than SES samples). This will tell us whether the gains from SES are largely due to the structured and process-supervised nature of the training data, or whether it is simply due to the fact that you are training on a larger number of tokens.

**Questions:**

- Have you tried combining SES with test-time search? i.e. actually generate the trees during test time and do search over them (rather than just doing it internally). This may yield further improvements and better scaling properties.

---

> ### Author Response · Authors · 2025-11-13
>
> Thank you for the thoughtful and constructive feedback. We genuinely appreciate both the recognition of our core contributions and the specific, actionable suggestions for improvement.
>
> **1. Failure Mode Analysis:** Almost all failures trace to hallucination within the reasoning trace itself. For instance, even when we explicitly stated a pit at (2, 3), the model sometimes treated it as passable mid-reasoning. We also observed rare cases (roughly 3 in 100 trials) where the model cycled through states repeatedly, saying "(1, 1) up → (1, 2) down → (1, 1) up..." indefinitely. This suggests the model hasn't fully internalized a systematic exploration protocol that prevents revisiting states. Detailed input-output formats are already in Appendix B across all tasks.
>
> **2. Task Difficulty and Scope:** We appreciate your suggestion to evaluate on MiniHack and Crafter, but we'd gently push back on the framing of our tasks as "simple." Humans and LLMs have fundamentally different notions of difficulty, and we think the empirical numbers tell an important story here. Game of 24 achieves only 2% accuracy on base models with zero-shot prompting. Our 20x20 Gridworld achieves 0% with few-shot prompting even with four-shot examples. These aren't toy problems; they're legitimately challenging for even the most capable models available.
>
> The key insight is that structured and verifiable solutions don't make problems easy. They make them good benchmarks for controlled study. In fact, that's precisely why we chose them. Gridworld requires planning up to 60 steps ahead to guarantee optimality, which is genuinely difficult for language models. When a model achieves only 0% with a few-shot prompt, we're not dealing with something trivial.
>
> Regarding MiniHack versus our setting: we're genuinely unsure if MiniHack is harder. While MiniHack does support offline planning, the world itself doesn't appear to be as expansive as ours. Our task demands offline planning across 50–60 future steps with combinatorial branching factors; to guarantee optimality, the model must systematically explore a genuinely vast search tree. MiniHack, though procedurally generated and feature-rich, may not necessarily require reasoning over such a large effective problem space. Crafter, on the other hand, has a large world, but the catch is that it fundamentally requires interaction and partial observability. You can't plan the entire trajectory offline because the observation changes as the agent moves. That's a different problem class from the offline planning scenario we're targeting, which makes it less of a direct comparison. Both remain interesting directions for future work.
>
> **3. Compute-Controlled Comparison:** Looking at Figure 9 in the appendix, we can estimate the total token budget ratio. The average input length is roughly 331.98 tokens, and the output lengths differ significantly: SES uses approximately 666.31 tokens for output, while direct SFT uses only 25.70 tokens. That gives us a total ratio of (331.98 + 666.31) / (331.98 + 25.70) ~= 2.79x.
>
> What this means in practice is: we extended training each model up to 3125 steps. An SFT baseline trained for 3125 steps achieves roughly 23% scaling success, while SES at 1250 steps (which is approximately 2.79x fewer tokens) achieves 82% scaling success. This is actually quite striking and suggests the advantage isn't simply due to training on more tokens.
>
> **4. Combining SES with Test-Time Search:** This is an interesting, but as of I understand about test-time scaling, there's a fundamental constraint that makes it tricky: our model only terminates when it explicitly outputs the answer token, so it’s naturally hard to control the token budget for test-time scaling.
>
> That said, there is exciting concurrent work exploring this space more directly. A related paper (https://openreview.net/forum?id=PV5Dy4lW3t) takes a complementary approach: instead of linearizing the entire search tree into one unified reasoning trace as we do, they "fork" each child node into separate branches within the reasoning. The key difference is that while they generate all children under the same parent within a single generation step (allowing them to produce diverse thinking fragments causally), they only concentrate on planning the current path they're generating, so they don't linearize the full tree. This is a genuinely different strategy and one we think is worth monitoring. It may ultimately be better suited to test-time scaling, since they can easily control the number of child nodes they want to generate.
>
> We view all of these points as opportunities to strengthen the paper and clarify our positioning. Your feedback has helped us see where we need to be more explicit. We're genuinely grateful for the engagement.

---

### Official Review · Reviewer_NiWi · 2025-10-31

**Soundness:** 3
**Presentation:** 3
**Contribution:** 2
**Rating:** 2
**Confidence:** 4

**Summary:**

The authors introduce a framework designed to teach language models a search operator. Instead of verbalizing only a single reasoning path, SES verbalizes complete multi-branch search traces.

The key components verbalized in a Systematic Exploration Chain-of-Thought (SE-CoT) trace are:
1. Sampling: Enumerating candidate actions systematically or in a reward-ordered sequence.
2. Propagation: Predicting resulting states or intermediate evaluations.
3. Backtracking: Recovering the optimal path from the goal to the start.

SES is implemented via two methods:
1. Direct Algorithmic Supervision: Used when a canonical search procedure (e.g., BFS/DFS) is known (e.g., textualized Gridworld). This method explicitly supervises the entire search process, including counterfactual sibling branches, teaching *why* non-chosen actions were rejected.
2. Bootstrapped Search Discovery: Used for open domains without explicit algorithms (e.g., Game of 24). This procedure constructs synthetic search trees guaranteed to include at least one valid solution (by injecting the gold path) while adding diverse, reward-prioritized alternatives sampled from the base LM ($\pi_{\theta}$).

When trained only on 10×10 Gridworld environments and tested on unseen 20×20 grids, SES maintains 76.5% success, achieving a remarkable 4x improvement over standard SFT (19.0%) and significantly surpassing Tree-of-Thought (26.0%) and GRPO (6.0%).

**Strengths:**

1. Written clearly, easy to understand
2. Experiments are documented well

**Weaknesses:**

1. Lack of novelty: implementing search procedures during CoT is not at all something new and has appeared as early as [1], and has been a repeated area of focus throughout the test-time scaling breakthroughs in o1, r1. "Verbalizing complete multi-branch search traces (sampling alternatives, propagating outcomes, and backtracking to extract a solution) rather than the winning trace" is an area that has been very, very well-explored. Also, the study of test time exploration at token budgets beyond the training regime has also been well-studied before [2]. These works should be cited and discussed in related works.
2. Easiness of tasks and old models: Game of 24, GSM8K are both well-saturated tasks. Qwen 2.5 family is also not the newest family of Qwen. It would be more interesting to look at more difficult tasks, such as AIME, USAMO, or ZebraBench.
3. Lack of gain on open-reasoning domains, such as GSM8K.

[1] https://arxiv.org/abs/2404.03683
[2] https://arxiv.org/abs/2506.09026

**Questions:**

1. Perhaps it would be more interesting to compare different ways to construct reasoning traces from scratch, and getting really weak base models model to fit on it as a way to learn to implement test-time search? One could study how one can best construct such traces, and what heuristics / intricacies one might have to care about during this process.
2. Can we evaluate on harder datasets, as mentioned in the section above?
3. To push the story further, better performance on open-reasoning domains should be shown.

---

> ### Author Response · Authors · 2025-11-12
>
> We appreciate the detailed feedback. However, before we address each of your concerns, we think there's a fundamental category confusion we need to clarify first, because we believe once we align on this, many of your points will resolve naturally.
>
> Your concern is that "Search traces with branching and backtracking have been explored in o1, r1, etc., so this is not novel." We absolutely agree that o1-like search traces aren't new. But that's actually not what we're claiming is novel. Our core contribution is answering **"How do we train on o1-like search traces offline, without massive RL compute or cold-start brittleness?"**
>
> That's a different question entirely.
>
> Let us be specific about the papers you mentioned. SoS modifies the training signal iteratively (similar to STaR), but it doesn't explicitly construct complete search trees in the training data. It rather focuses on iterative refinement, not structural supervision. E3 tackles the RL brittleness problem directly, but it still relies on RL-based methods. Our contribution is orthogonal: we're proposing offline SFT as an alternative approach for scenarios where RL infrastructure is infeasible.
>
> What SES uniquely does (and which none of the previous works addresses including SoS and E3) is a combination of three things. First, we explicitly construct complete multi-branch search trees in the training data using both systematic and bootstrapped algorithms. Second, we guarantee that valid solutions are included while simultaneously adding diverse, reward-prioritized alternatives. Third, we train entirely offline via standard SFT with no cold-start problem and no parallel test-time search overhead. And critically, we evaluate this specifically for complexity extrapolation; That is, training on 10x10 grids and testing on 20x20.
>
> The novelty isn't in search traces themselves, but more in the **offline bootstrapped construction methodology combined with SFT training**, which is a concrete way to teach internalized exploration without RL.
>
> So we have three concrete questions for you: First, does any existing work show how to construct complete, valid search trees without requiring an explicit heuristic algorithm (like our Algorithm 2 does)? Second, do they train on these trees entirely offline via SFT, without any brittleness of RL? Third, do they demonstrate complexity extrapolation, by training on simpler instances and generalizing to harder ones?
>
> If the answer to all three is no, then we believe our methodology genuinely is novel. If there's a paper that does all of this, we're genuinely interested to see it. So if you know of one, please point us to it.

---

### Author Response · Authors · 2025-11-13
**Clarification on algorithm implementation**

We want to clarify the missing part of the algorithm that is not clearly posted in the current submission (thanks for pointing out, reviewer puAH!).

**Backtrack, Propagation, and Sampling** are illustrated in Figure 1, which shows the complete systematic exploration process: (a) Sampling shows systematic exploration of possible actions, (b) Propagation shows predicting outcomes for each action, and (c) Backtracking shows identifying the optimal path by reversing from goal to start.

The action and state definitions are task-dependent. For Gridworld, an action is a {state, action} pair where actions are {up, down, left, right}. For Game of 24, an action is selecting two numbers plus an operation. For open-ended tasks like GSM8K and ProntoQA, an action is the next reasoning sentence. You can see the concrete input-output formats for all these in Appendix B

**GetGoldAction** in Algorithm 2 means: given the current state in the search tree and the known gold solution trajectory(which is denoted as **GetGoldSolution**), return the next gold action that should be taken according to that trajectory. For example, the gold action for Step 1 in Appendix B.2.3 is "It takes 2 ∗ 0.5 =<< 2 ∗ 0.5 = 1 >> 1 bolt of white fiber.". For Step 2, it's "So it takes 2 + 1 =<< 2 + 1 = 3 >> 3 bolts in total.", and it goes on.

**Verbalize** converts the search tree structure into the text format shown in Appendix B—essentially linearizing the tree breadth-first with appropriate separators(ex. [Sep1] for dividing sequential children nodes, [Sep2] for dead-end marking, and [Sep3] for the answer signal)

We should have made these connections more explicit in the algorithm descriptions. In the revision, we'll add explicit references so readers can immediately connect the pseudocode to the concrete implementations.

---

### Meta-Review · Area_Chair_4eYq · 2026-01-07

**Summary:**

The authors introduce a framework designed to teach language models a search operator. Instead of verbalizing only a single reasoning path, SES verbalizes complete multi-branch search traces.

The method is very similar to SoS, and note that SoS does run SFT experiments, and so do many other papers beyond it. No comparisons, conceptual or empirical, have been made. The authors claim the paper is not about generating search traces but rather it tackles the question: "How do we train on o1-like search traces offline, without massive RL compute or cold-start brittleness?". It is not clear to me which ablations prove that their design is good or solves this problem more generally if the study is on a gridworld/Game of 24, other benchmarks such as GM8K show no gains, and no other more difficult benchmark has been studied (reviewers have suggestions for these). More importantly, not even a single other approach of doing SFT is compared with. Overall, due to this reason I don't think the paper is significant (or the significant contribution is not explained properly) and should be rejected at this time. Please take reviewers' feedback into account for revising the paper.

**Reviewer Concerns:**

Lack of novelty, imperfect clarification of contributions, improper baselines, no ablations

**Reviewer Scores:**

All concerns remain -- my guess is the reviewer with a score of 8 would have reduced down to 4 after reviewer discussion.

---

### Decision · Program_Chairs · 2026-01-26

Reject